# Fast cycling culture of the annelid model *Platynereis dumerilii*

**Mathieu Legras**[1¤], **Giulia Ghisleni**[1,2], **Léna Regnard**[1], **Manon Dias**[1], **Rabouant Soilihi**[1], **Enzo Celmar**[1], **Guillaume Balavoine**[1]*

**1** Université de Paris Cité, CNRS, Institut Jacques Monod, Paris, France, **2** Department of Biotechnology and Biosciences, University of Milano-Bicocca, Milano, Italy

¤ Current address: Université Paris Saclay, CNRS, Institut des Neurosciences, Saclay, France
* guillaume.balavoine@ijm.fr

**Data Availability Statement:** All data files are available at the Dryad repository https://datadryad.org/stash/share/S0IXlKFfdfxmxyHLJUWEQvoq9vIum7bQDuX3nRGPHBw.

**Funding:** The Balavoine group was financially supported by the CNRS, the University Paris Cité,

## Abstract

*Platynereis dumerilii*, a marine annelid, is a model animal that has gained popularity in various fields such as developmental biology, biological rhythms, nervous system organization and physiology, behaviour, reproductive biology, and epigenetic regulation. The transparency of *P. dumerilii* tissues at all developmental stages makes it easy to perform live microscopic imaging of all cell types. In addition, the slow-evolving genome of *P. dumerilii* and its phylogenetic position as a representative of the vast branch of Lophotrochozoans add to its evolutionary significance. Although *P. dumerilii* is amenable to transgenesis and CRISPR-Cas9 knockouts, its relatively long and indefinite life cycle, as well as its semelparous reproduction have been hindrances to its adoption as a reverse genetics model. To overcome this limitation, an adapted culturing method has been developed allowing much faster life cycling, with median reproductive age at 13–14 weeks instead of 25–35 weeks using the traditional protocol. A low worm density in boxes and a strictly controlled feeding regime are important factors for the rapid growth and health of the worms. This culture method has several advantages, such as being much more compact, not requiring air bubbling or an artificial moonlight regime for synchronized sexual maturation and necessitating only limited water change. A full protocol for worm care and handling is provided.

## Introduction

During the process of experimental design, the choice of the most appropriate model organism is a crucial step that determines the validity of the study. Gregor Mendel, the pioneering author of some of the most famous genetics experiments on plants of the genus *Pisum*, stated that the selection of an unsuitable model can make the results questionable from the outset [1]. The pea plant was not only physiologically and morphologically suitable for his "Experiments on Plant Hybrids", but its representational power has made it a model organism by definition [2, 3]. Knowledge derived from a model organism can be projected and generalized to a broader range of systems, and this, together with the integration of multidisciplinary approaches, allows for a 360-degree comparative study between species across the tree of life. For example,

the Institut Jacques Monod, the Agence Nationale de la Recherche (grant PRCI TELOBLAST ANR-16-CE91-0007) and the Fondation ARC pour la recherche sur le cancer (grant LSP 190375). GG was supported by a Master student fellowship of the EUR G.E.N.E. graduate school (#ANR-17-EURE-0013) that is part of the Université Paris Cité IdEx #ANR-18-IDEX-0001 funded by the French Government through its "Investments for the Future" program.

**Competing interests:** The authors have declared that no competing interests exist.

in the case of bilaterian animals, studying mice, chickens, fish, and sea urchins (which belong to the deuterostomes) together with *C. elegans* and *D. melanogaster* (ecdysozoans) has enabled a comparative approach to regulatory cladistics that has revealed the shared nature of signalling pathways and transcription factors among all modern Bilaterians [4]. The evolution of phylum-specific body plans and morphological structures is generally not due to the rise of new developmental genes, but we can expect and predict that novel regulation circuits have developed [5]. Investigating how developmental genes are regulated in different lineages can therefore provide new evolutionary insights into the beginnings of the history of Bilaterians.

In this context, the marine annelid *Platynereis dumerilii* [6] compensates for the longstanding lack of representatives of the lophotrochozoans, which is the third branch of bilaterians alongside deuterostomes and ecdysozoans, that has impeded a reliable approach to bilaterian comparative genetics. Carl Hauenschild was the first scientist to begin culturing *P. dumerilii* in 1953 and standardize its culture conditions [7] referred to henceforth as "traditional." The potential of *P. dumerilii* as an emerging animal model lies in the study of its molecular and developmental processes. It is considered to have retained not only most of the ancestral traits of polychaete annelids but also many ancestral characteristics inherited from *Urbilateria*, the last common ancestor of all Bilaterians [8, 9]. Earthworms and leeches have been used as lophotrochozoan animal models [10, 11], but they show derived features within the annelid clade. They lack many of the head structures and trunk appendages, and their life cycle lacks a larval phase and metamorphosis, which are ancestral developmental traits of annelids [12] and likely of bilaterians [13]. In contrast, all these characteristics are present in the polychaete family Nereididae, to which *P. dumerilii* belongs, making it a suitable and strategic model for providing insight into the deep morphological and developmental past of bilaterians.

*P. dumerilii* is found in all European seas [14]. When managing hundreds or thousands of individuals, the fact that they are undemanding, small-sized, and easily manipulated is crucial. The ability to culture *P. dumerilii* in the laboratory for the full life cycle, its rapid and highly synchronized embryogenesis, and the transparency of embryos and juveniles allowing whole body [15] and live imaging [16], represent a significant advantage for studying reproductive biology, development, and regeneration.

The developmental stages of the life cycle have been extensively described [7, 17–19]. The following description is a summary detailing stages observed at a culturing temperature of 18˚C (Fig 1). A single mating of *P. dumerilii* yields thousands of fertilized eggs. These eggs undergo synchronized development into larval stages. The zygote contains lipid droplets, cortical granules, protein yolk granules, and secrete a protective jelly. Abundant jelly secretion indicates successful fertilization. During the early stages of *P. dumerilii* development, embryos exhibit stereotypical spiral cleavage [20]. The initial divisions are highly asymmetric, and cells can be identified based on their position and size. During this spiral phase, the fusion of lipid droplets forms four large ones, indicative of healthy development. Larvae with more or less than four lipid droplets will not develop normally.

By 24 hours post-fertilization (hpf), the trochophore larvae are free from their jelly, start swimming with the help of a ciliary band, and exhibit positive phototaxis. At this stage, larvae are planktonic and do not feed. The only nutrient source is the yolk. By 4 days post-fertilization (dpf), the nectochaete larvae have produced a head, a pygidium (posterior end), tentacular cirri, anal cirri, and three chaetigerous segments with parapodia. The larvae then move from the pelagic zone to the benthic zone and spin a tube made of a mixture of silky fibers and glue (rather than mucus, as occasionally mentioned) [21], in which they will spend most of their lives. At 6 dpf, they begin feeding (mostly at night when they leave the tube) and add new segments to the posterior end from the segment addition zone (SAZ), located just anterior to the pygidium [8, 18]. From now on, the developmental pace of juveniles will vary between

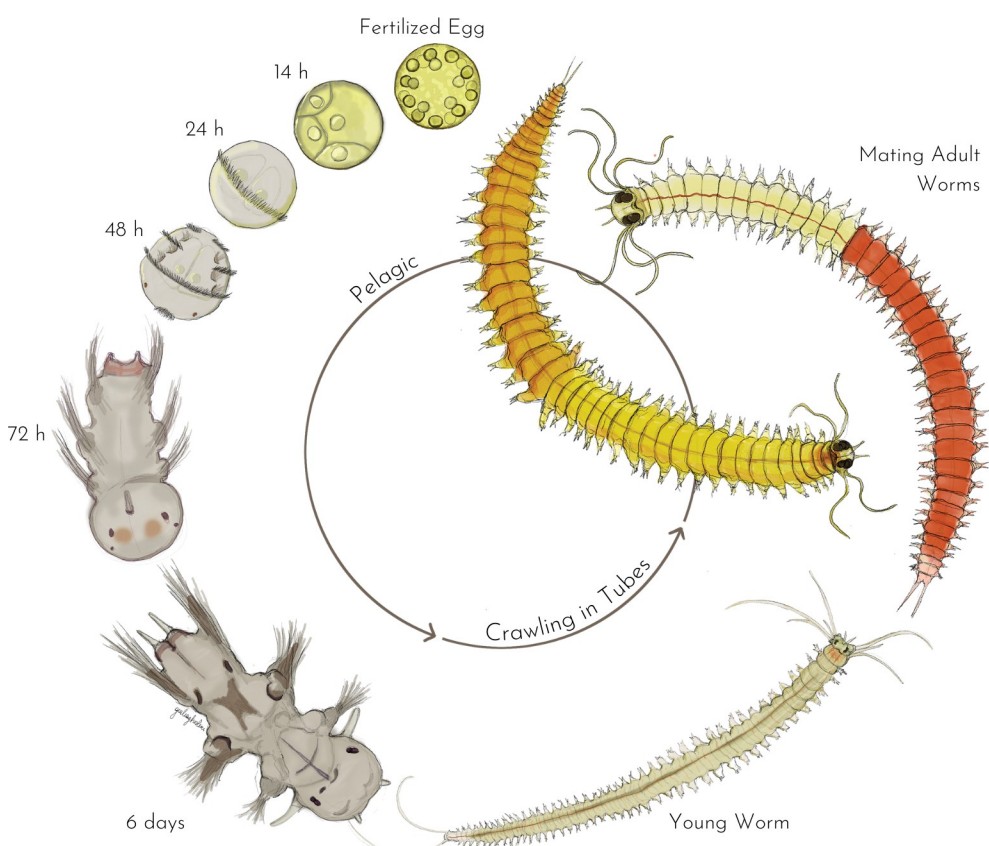

**Fig 1. The life cycle of *Platynereis dumerilii*.** The various life stages are depicted and classified based on the worm's pelagic/benthic behavior. The mentioned time points correspond to those obtained at a thermostated temperature of 18°C [18]. The organism has a diameter of approximately 160 μm from the fertilized egg to 48 hpf. By 72hpf, the swimming larva measures around 250 μm in length. Around 6 dpf, upon starting feeding, it begins elongating through posterior segment addition. Immature worms reach 3–5 cm length. At the end of sexual metamorphosis, the worms become bulkier, contracting longitudinally by almost 50%.

individuals. The size of the worm depends on its food regime and quantity of food intake. Before sexual maturation, gonial clusters begin to populate the body of the worms when they reach around 40 segments [22] and gametes start to fill the coelom at approximately 50 segments. Worms eventually become mature at ~70–80 segments [23, 24]. During sexual metamorphosis, the worms stop feeding and their gut degenerates. Their eyes increase in size and extensive changes occur in their muscles, parapodia, and body color, as they transition from a sexually immature atoke form to a sexually mature epitoke form [7]. Sexually mature worms need to swim fast, so their muscles degenerate to make room for a new epitokous muscle type. Oocytes are yellow and give this color to mature females. Males are bicolor due to the white sperm that colors the anterior part of the worm, while the posterior part is red due to blood capillaries filled with hemoglobin [25]. Once epitoky is complete, spawning of *P. dumerilii* peaks around 1 week after the full moon phase [26]. The worms leave their tubes and move into pelagic water to look for a partner. Triggered by pheromones [27, 28], the mating couple starts a nuptial dance (circular fast swimming) and the female releases all oocytes into the water. The male fertilizes them externally and the worms die shortly after the release of gametes. All timings of these developmental stages are strongly influenced by changes in temperature [18].

Several studies have demonstrated the suitability of *P. dumerilii* for reverse genetics experiments, including transgenesis using DNA transposase [29, 30] and targeted gene knockouts with TALENs or Cas9 [31, 32]. However, despite its potential as a genetic model organism, there are still challenges to making *P. dumerilii* a widely used "fruit fly of the sea." One of these challenges is the extensive genetic polymorphism present in laboratory strains, even after decades of maintenance in various European and American laboratories. Non-inbred strains exhibit a high proportion of SNPs (Single Nucleotide Polymorphism) [33]. This is particularly relevant to genome editing by CRISPR-Cas9, as a PCR polymorphism study must be conducted for each targeted gene [32]. Furthermore, reproduction in *P. dumerilii* occurs only once at the end of the life cycle, and adults die immediately after spawning. This makes it difficult to maintain homozygous strains, as adults of both sexes must be obtained on the same day, requiring large batches of transgenic animals to be produced and maintained until they reach sexual maturity.

The main challenge remains the relatively long reproductive cycle of *P. dumerilii*. In nature, the reproduction period is seasonal, implying that most worms live for about a year. In the laboratory, temperature (generally 18˚C) and light regimes (8/16 hrs of night/day) are used to simulate the end of springtime. To replicate the role of moonlight in synchronizing the swarming of male and female adults, an artificial moon, in the form of a small bulb or LED strip [34], is placed in the culture room for one out of four weeks. These conditions, combined with regular feeding, can induce sexual maturation at a much younger age. However, the age of reproduction remains highly variable and is not correlated between worms from the same parental batch. A batch of siblings, under the traditional culture protocol, will usually produce adults whose ages vary between 4 and 12 months, making it difficult to create and maintain transgenic strains.

We hypothesize that three factors contribute to the variability in sexual maturation age. Firstly, the highly variable density of worms in culture boxes may be a factor. *P. dumerilii* juveniles are typically kept in flat plastic alimentary boxes with a bottom surface area of around 500 cm$^2$. Young worms weave tubes at the bottom of the box in which they spend most of the day. Instead of being randomly distributed, the tubes are usually built in positions that maximize distances between worms, suggesting territorial behaviour (S1B Fig). This behaviour is possibly induced by antagonistic interactions between juvenile worms, as wandering worms are frequently attacked and bitten by their neighbours. Secondly, the quantity of food delivered is not adapted to the number of worms in the boxes. Too little food results in slow growth, while too much food can lead to water fouling with the same effect or even worm death by asphyxia. In traditional culture methods, water must be changed every two weeks to tackle fouling [7]. Finally, the extensive genetic polymorphism observed in our historical culture population (which we call the polymorphic strain) and verified in numerous PCR studies on various genes has led us to hypothesize that genetic factors may also be involved in the variability of maturation age.

In this article, we describe how we have established a new protocol for culturing a selected strain of *P. dumerilii* called Fast Forward (FF) by controlling density, adapting food delivery, and selecting worms based on their maturation age. Sexual maturation for this strain begins at 10–11 weeks, and most worms mature before 18 weeks with an average maturation age at 13.2 weeks. This protocol places *P. dumerilii* among the small minority of animal models in which transgenic and genome editing techniques can be developed with ease in the future. Most importantly, we have significantly simplified the culture conditions by eliminating the need for a moon cycle and reducing the frequency of water changes, thereby reducing staffing requirements and enabling the rearing of several transgenic strains in limited lab space.

## Materials and methods

All Platynereis worms used in this study are originating from a culture established in the institut Jacques Monod since 2009. These worms are referred to as the "polymorphic strain". They are derived from the original culture set up by Carl Hauenschild in Germany 70 years ago. Animal research in the European Union (EU) is regulated under Directive 2010/63/EU on the protection of animals used for scientific purposes. The annelid *Platynereis*, as other invertebrates except cephalopods, is not covered by this regulation and no authorization is required for experimentation.

### Making new batches of larvae

Mature worms are collected from low density boxes every morning. Male worms display a bicoloured white/red pattern, whereas female worms are yellow/orange. Only actively swimming adults are chosen daily for reproduction. To make all fertilizations and keep the embryos/larvae up to ten days after fertilization, small beakers (diameter 9.5 cm, height 5.5 cm) are used. Approximately 150 ml of natural filtered seawater (NFSW) is added in each beaker, along with one pair of worms (Fig 2). All worms are manipulated using plastic pipettes (Samco scientific). Ideally, the release of gametes occurs in the next few minutes. Sometimes, this release does not happen spontaneously even though both individuals are active. In such cases, spawning can be achieved by gently pressing the male with a fine paintbrush to force the release of a small amount of sperm. This, in turn, triggers the release of the female oocytes. Once both adults have released their gametes, most of the NFSW used for fertilization must be poured out to prevent overnight fouling by the male sperm. This is done easily since the fertilized eggs sediment quickly at the bottom of the beaker. A new volume of 150 ml of clean NFSW is then added. The efficiency of the fertilization is checked by the formation of a transparent jelly coat around the eggs. Thirty minutes post fertilization, the eggs adopt a characteristic hexagonal spatial distribution while being pushed apart by the jelly. To prevent fouling of the water due to the decomposition of the jelly coat, 1.5 ml of 100x antibiotic mix is added. Fertilized batches are incubated overnight at 20˚C. The next day, at 24 hours post-fertilization (24 hpf), the larvae must be dejellified (getting rid of the jelly coat). The content of the beaker is poured into a large 80 μm sieve (S2 File), allowing the jelly to sieve through with the seawater. After two quick rinses with NFSW, the sieve is returned upside down to the original beaker and 150 ml NFSW is poured over the whole sieve to recover all swimming larvae. Small worms are incubated at 20˚C until they reach 4 dpf. They are then fed with 3 ml of 1x frozen microalgae (*Tetraselmis marina*, Instant Algae™, Table 1). Three-segment small worms should start to feed and settle at the bottom of the beaker quickly.

### Transplanting new boxes of juvenile worms

To achieve maximum survival of juvenile worms, it is important to use healthy batches of young worms. At 10 dpf, the beaker containing young worms should consist primarily of feeding individuals, which can be checked by a gut full of algae and the budding of a fourth segment in some. However, batches of eggs show significant variability, as indicated in the troubleshooting guide. Numerous batches contain larvae with abnormal development. Subsequently, many batches exhibit a proportion of juveniles that, despite being provided with algae at 4 days post-fertilization (dpf), fail to feed by 10 dpf. Typically, these juveniles with empty guts will not progress in development and will eventually die. For further culturing, we will use healthy and rapidly growing 4-segment young worms and avoid 3-segment worms and those with an empty gut. The boxes used for the main phase of growth are polypropylene containers, that are 6.5 cm high, and can be easily stacked in incubators or on the bench. These containers

**Fig 2. Protocol overview for Fast Forward strain culturing in *Platynereis dumerilii*.** The Results section provides a detailed description of the overall progression towards selecting FF individuals and achieving the proposed protocol.

**Table 1. Weekly schedule for the distribution of food in individual boxes.**

| Worm ages | Monday | Wednesday | Friday |
|---|---|---|---|
| 4–30 dpf | 3 ml algae | | 3ml algae |
| 31–45 dpf | 1ml Sera micron® | 3 ml spinach | 1ml Sera micron® |
| > 46 dpf | 3ml Tetramin® | 3ml spinach | 3ml Tetramin® |

All volumes are suspensions in NFSW at the concentration specified in the text. Worm ages are in days post fertilization (dpf).

have lids that are not completely airtight, ensuring vital gas exchange for the worms. Typically, four boxes are transplanted from each batch of young worms. Each box is filled with 500 ml of NFSW and supplemented with 3ml of 1x frozen algae and one cm$^2$ of old box algal mat (see commensals section). 25 young worms are transplanted into each box, achieving the optimal density of 375 individuals.m$^{-2}$ (as determined in this study). Young worms can be selected using a P20 hand pipette with bevelled pipette tips. Worms reflexively grip the inside of the tip with their bristled appendages, so it is important to set the hand pipette at a very small volume (1 µl) to pipette individual worms and expel them out of the tip efficiently. The boxes are incubated at 20°C, with a 16-hour day / 8-hour night light regime, for the entire remaining life cycle.

## Food regime

The food regime is inspired from the traditional culture system [7]. Four different types of food are used successively to ensure rapid and healthy growth. The quantities mentioned must be strictly followed, as overfeeding can cause water fouling and be a major cause of death for juvenile worms. The preparations of all food suspensions are described in S2 File. The microalgae typically used for feeding early juvenile stages are live *Tetraselmis marina*. To avoid time consuming lab culture of this green alga, commercial fish hatchery food (Instant Algae™, Reed Mariculture) is used. The algae are diluted 40 times in NFSW, aliquoted in 50 ml Falcon tubes and kept frozen at -20°C. Sera Micron® (Sera) is the food chosen for older juveniles, as its particle diameter is well suited for the worm's mouth at this stage, and it is highly nutritious for rapid growth. It is resuspended (1% weight/volume) in NFSW and must be kept frozen if not used immediately. Tetramin® flakes are fed to worms older than 46 days. They are ground to a powder using a mortar and are suspended in NFSW (1% weight/volume). Tetramin® suspension can be kept frozen if not used on the same day. Finally, frozen organic spinach is used to supplement both Sera® and Tetramin® regimes, in a 10% weight/volume suspension that is ground with a food blender. Food distribution is made according to Table 1.

## Worm growth monitoring and water change

Young worms are not visible to the naked eye during the first three weeks of culturing in polypropylene boxes. After 3–4 weeks, they start spinning tubes that are large enough to be observed with the naked eye at the bottom of the containers. Therefore, it is recommended to check after one month that growth is normal, and no mortality of young worms has occurred. At two months post fertilization (61 dpf or 2 mpf), a water change is necessary. At this age, worms are firmly settled in their tubes during the day and the old water can be safely poured out into an intermediate container before being discarded. Occasional runaway worms can be put back into their box with a plastic Pasteur Pipette. 500 ml of fresh NFSW is added to each box. Worms are counted at this 2 mpf stage to ensure that there are still between 20 and 25 worms in each box. As the worms will sexually mature rapidly over a period of 6–7 weeks

starting at 10–11 weeks, no further water change will be required. As the worm counts decrease in each box through maturation, boxes containing five remaining worms or fewer are regrouped to save space in the incubator. To handle worms out of their box, plastic Pasteur pipettes are used. Worms are chased from their tube by gently compressing the tube with the pipette tip, starting at the end of the tube where the head of the worm is facing. In this way, the worm wiggles backward out of the tube, reducing the risk of occasional injury and sectioning of the fragile tail.

## Commensals

*P. dumerilii* cultures are anything but axenic. Boxes contain several other organisms, some of which are likely beneficial for the growth and maintenance of the annelid, while others may grow to the point of interfering with the well-being of the worms. These co-cultured organisms are typically passed down from generation to generation when transplanting worms to new boxes with pipettes and are not detected due to their microscopic size. Alternatively, they also come with the natural seawater (in our case, natural filtered seawater from the bay of Cancale in Brittany), which, despite filtering, is not fully sterile. It is important to mention that we have never observed cases of internal parasitism, such as by myxozoans [35], in our culture. This indicates that the filtration quality of the sea water we used has been satisfactory in this respect. In this section, we will give a brief description of these commensals. We will describe how to transfer them from boxes to boxes if they may be a complementary source of food for the worms, help clean out excess distributed food, or help maintain the homeostasis of the box in any other respect. We will also describe how to get rid of unwanted commensals.

In the older boxes, the bottom gradually becomes covered with green filamentous algae and cyanobacteria mats (Fig 3, S2C Fig). Small worms feed on filamentous algae, as the algae gets cleared near the extremities of the worm tubes. Additionally, these photosynthetic organisms are likely important in maintaining a satisfactory level of dissolved oxygen in the water, as we never place air diffusers at any stage of this culture. We have observed that small worms settle in tubes earlier in boxes where we ensure that filamentous algae and other components of the mat grow rapidly. To obtain quick growth of the mat, we inoculate boxes with small pieces of the mat from a 4-month-old box. It is important to ensure that this old box does not contain any unwanted commensals (such as *Dimorphilus* worms, see below) or small *P. dumerilii* larvae or tiny juveniles from spontaneous reproduction, which can occur even when boxes are checked for adults daily. The mat is scraped from the box bottom using a plastic scraper. A small piece roughly 1 cm$^2$ is sufficient to inoculate a new box. This is done at the same time as new boxes are populated with 10-day worms (Fig 2).

Unicellular eukaryotes are also present in our culture boxes, consisting of two groups: hypotrich ciliates wandering on the box bottom and swimming flagellated unicells (mostly dinoflagellates). In boxes, their proliferation is harmless to the worms and may even benefit the culture by cleaning out excess food. These unicellular eukaryotes are propagated from one generation of boxes to the next by pipetting adults or seeding filamentous algae, as practiced in the lab. These organisms do not require extensive monitoring, except for one exception. In the beakers containing larvae and small worms fed with *Tetraselmis*, dinoflagellates can proliferate excessively, rendering the water cloudy, depleting the medium of oxygen, and ultimately killing the small worms. If the recommended quantity of food and culture time in these small beakers with a high density of worms is followed (no more than 10 days), this problem is unlikely to become uncontrollable.

Lastly, small metazoans are sometimes co-cultured with *P. dumerilii*. Nematodes have been observed on several occasions, but they never reach high densities and do not affect the wellbeing of *P. dumerilii*. However, more concerning are the proliferations of the tiny annelid *Dimorphilus gyrociliatus* [36]. These limbless, gliding annelids, which are less than 1 mm long, can be

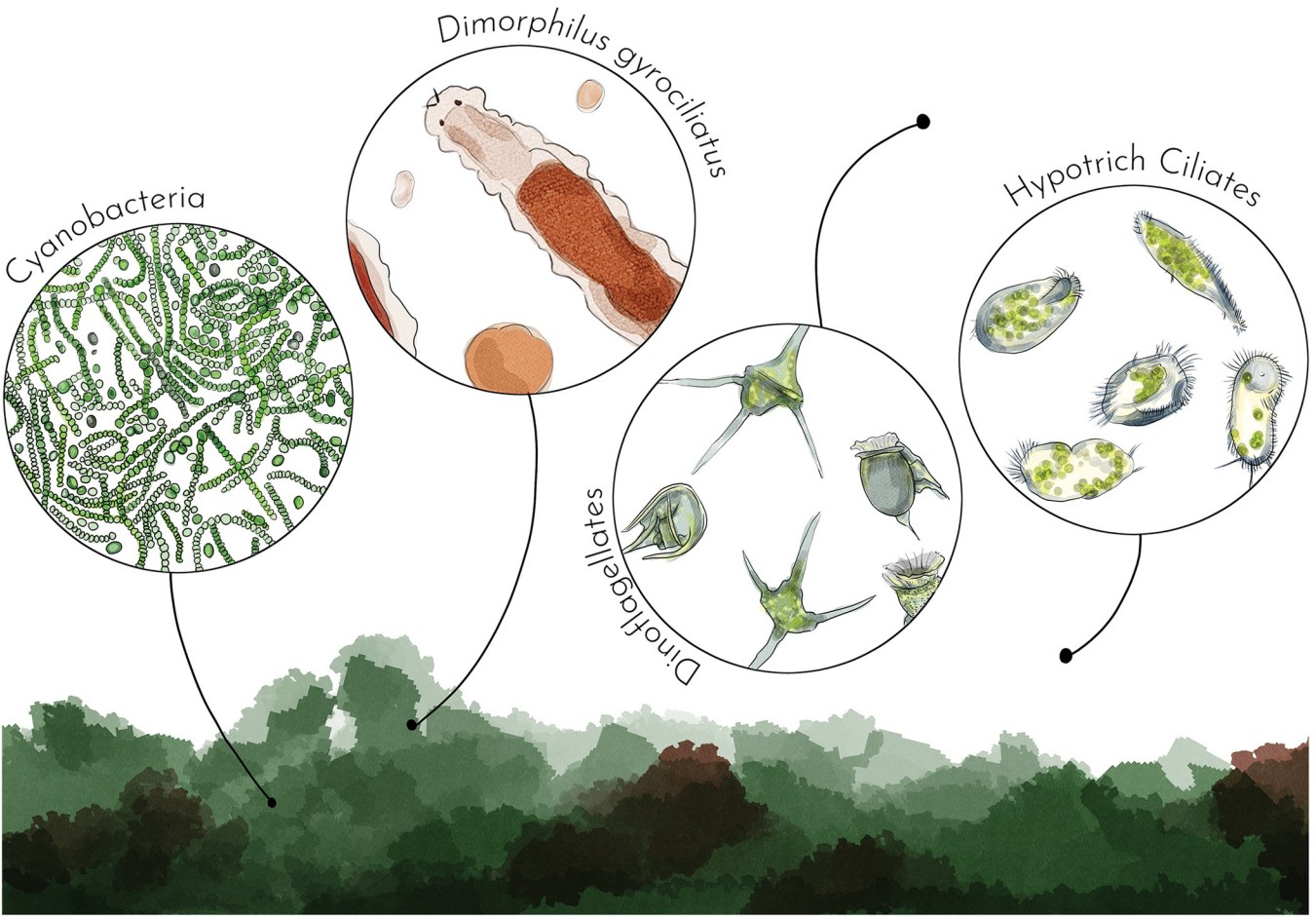

**Fig 3. Commensals commonly found in *Platynereis dumerilii*'s culture boxes.** Cyanobacteria and filamentous green algae are constitutive of the mat. *Dimorphilus*, nematodes and ciliates feed at the surface of the mat. Dinoflagellates are free swimming above the mat.

detected by examining the box bottoms with a stereomicroscope. In some of our low-density culture boxes, they have reached enormous densities, hindering the growth of *P. dumerilii* and potentially causing their death. To eliminate them, we transferred all *P. dumerilii* individuals to a new box. Each worm had to be rinsed with seawater several times to ensure no *D. gyrociliatus* remained. It is also important to follow simple rules to eliminate these unwanted commensals from the culture: the beakers containing the small 10-day worms used to populate new boxes should be checked for any contaminating *D. gyrociliatus*, and plastic pipettes used to collect adults should be discarded and replaced when collecting worms from boxes of a new generation.

## Data analyses and graphs

All analyses were performed in 'R 4.1.3' and RStudio. Graphs were created using the R package 'ggplot2 v3.4.1'. ChatGPT 3.5 was used for occasional help in R scripting.

## Results

Thanks to a gradual integrated approach in raising new generations of worms with new conditions of temperature, feeding and density, we were able to lower considerably the maturation

age. We then conducted control culture experiments to test the separate contributions of density and artificial selection on the maturation age.

## Artificial selection did not improve the average maturation age

An initial selection of four pairs of early-reproducing worms was made from the polymorphic stock. The worms obtained from these four pairs were called FF1G (1st generation). All crosses were made between worms of the same generation, which were further referred to as FF2G, FF3G, and so on. The dates of emergence of all mature worms were systematically recorded in each generation up to 130 days after fertilization. Only the earliest pairs of mature worms of each generation were retained for spawning the next generation. Importantly, we systematically avoided brother/sister crosses. The rationale for this restriction is to preserve as much of the initial polymorphism as possible to prevent the co-selection of deleterious mutations and the rapid random selection of sub-optimal combinations of genes in terms of maturation age. Another crucial factor in this process was the selection of fast-growing juvenile worms from 10 dpf batches to populate low-density boxes (see Materials and Methods). From FF5G onward, we exclusively selected the 10-day post-fertilization worms that had developed a fourth segment after six days of feeding with microalgae (Fig 2). This is a second level of selection that may have influenced the final gene frequencies of the FF population. Eight generations were obtained in this way over a two-year period. Excluding the problem that affected FF4G, likely due to the food regime (see next section), the median age of maturing worms decreased from generation to generation (Fig 4) but did not improve after FF6G. Because the process of elaborating this new protocol involve important changes in the food regime and the density of worms transplanted, the role of artificial selection in lowering the maturation age compared to the polymorphic strain could be experimentally assessed only when a stable protocol was properly determined and tested (FF6G-FF8G).

To determine whether artificial selection plays a role in fast cycling, we raised animals from the polymorphic stock using the same density and food conditions as those used in the last three generations of selected worms (FF6G-8G) and compared it to worms of the 10th generation of FF. In the "poly" control, worms were taken randomly at 4 dpf before any feeding to avoid any selection as fast-growing worms. The maturation curve of the polymorphic worms (Fig 2,"poly") does not appear significantly different from that of the selected worms FF6G-8G or FF10G, with a roughly equal median maturation time. We added a polymorphic control including a selection of fast growers at 10 dpf, as done for the FF strain. The median age of maturation for these polymorphic fast growers was the lowest obtained in the study, equal to FF6G (Fig 4). The maturation frequency of animals over time for the polymorphic (poly_4s) and first generations of selected worms (FF1G-2G) displays two peaks at 4-week intervals (Table 2), even without the application of an artificial moonlight regime (S1 Fig). Single waves of maturation are then observed in selected worms (FF3G-8G). The bimodal maturation in unselected worms could be linked to an intrinsic circalunar maturation rhythm [26], but this is merely a hypothesis. The overall sex ratio in selected worms was not significantly different from parity (Table 2) but there is a significant tendency for earlier male maturation (S1 Fig and Table 2), resulting in a slight excess of males in the first 3–4 weeks of mature collection. This discrepancy was not as sharply displayed in the control experiment, suggesting that it might be the result of artificial selection. This is again a hypothesis as the determinism of sex in *Platynereis* is not currently known. The earliest maturing individuals were 60 days old but did not produce viable offspring. The earliest mature animals involved in successful reproduction were 68–70 days old. Between 7 and 10 viable batches of eggs were selected for beginning the next generation of FF worms over the course of two weeks. The average age of the parents

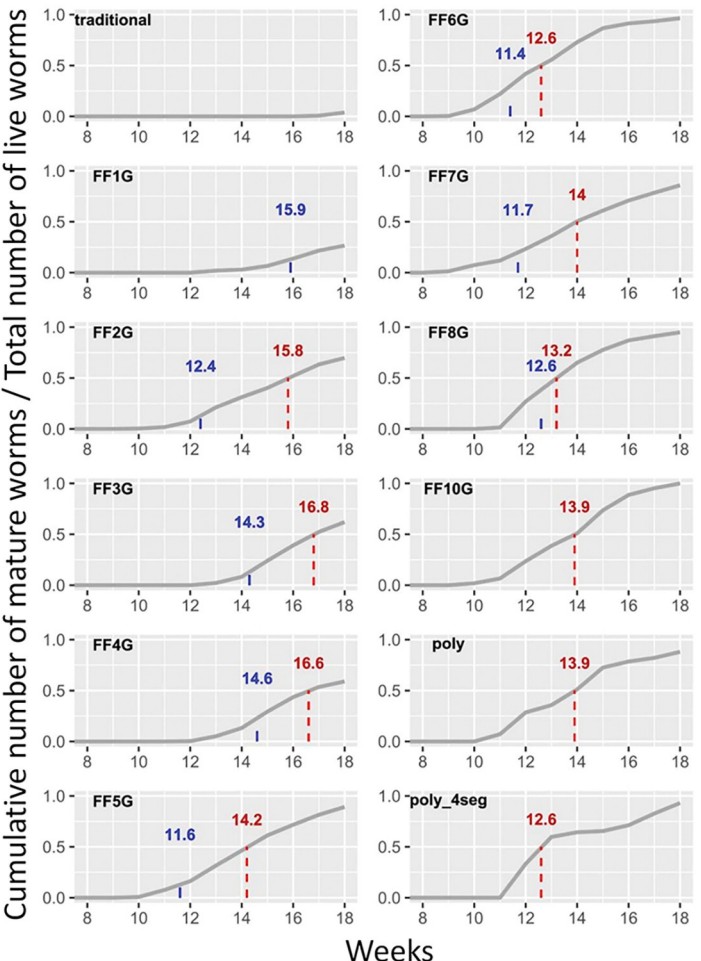

**Fig 4. Cumulative plots of mature worm count in each generation of the FF strain selection.** The columns are named as in Table 2. The density, food regime, and general statistics for each generation are described in Table 2. The dashed lines and red numbers indicate when 50% of the initially transplanted worms have matured (median maturation age). Blue numbers indicate the average age of worms selected to spawn the entire next generation.

used for these maintenance reproductions decreased rapidly (Table 2 and Fig 4 in blue) and stabilized around 12 weeks (FF6G-8G).

We modified the density and food regime of the worm boxes over the two-year period, and unlike artificial selection, these two parameters appeared crucial in maintaining a fast-cycling culture, as explained below.

## Feeding and water change

The traditional food regime utilized for our polymorphic *P. dumerilii* culture is based on the methods developed by Hauenschild and Fischer [7]. Initially, we provided *Tetraselmis marina* microalgae obtained from a lab culture at the Institut Jacques Monod (courtesy of P. Kerner) to little worms aged from 4 days to 1 month old. This was followed by a suspension of Sera micron® for the second month, and then a mixed regime of Tetramin® and organic spinach from the third month onwards. These various food items are adapted to the mouth size of the fast-growing worms. To simplify the procedure, we first tried to eliminate the organic spinach. Table 2 provides a timeline of the successive regimes and quantities of food delivered to the selected worms.

**Table 2. Main statistics for the successive generations of the FF strain selection.**

| | traditional | FF1G | FF2G | FF3G | FF4G | FF5G | FF6G | FF7G | FF8G | FF10G | poly | poly_4s |
|---|---|---|---|---|---|---|---|---|---|---|---|---|
| food regime | A | A | A | A | A | B | C | C | C | C | C | C |
| temperature (˚C) | 18 | 20 | 20 | 20 | 20 | 20 | 20 | 20 | 20 | 20 | 20 | 20 |
| water change interval (months) | 0.5 | none | none | none | none | 2 | 2 | 2 | 2 | 2 | 2 | 2 |
| artificial selection at 10dpf | none | none | none | none | none | yes | yes | yes | yes | yes | none | yes |
| number of worms | 132 | 406 | 565 | 963 | 593 | 638 | 385 | 623 | 326 | 140 | 145 | 100 |
| total matures collected < 130d | 5 | 125 | 372 | 582 | 187 | 466 | 268 | 410 | 208 | 106 | 78 | 82 |
| worm remaining > 130d | N/A | N/A | 147 | 295 | 123 | 43 | 9 | 36 | 7 | 0 | 6 | 5 |
| dead worms | N/A | N/A | 46 | 86 | 283 | 129 | 108 | 177 | 111 | 34 | 61 | 13 |
| number of boxes | 3 | 14 | 23 | 36 | 26 | 36 | 27 | 32 | 18 | 7 | 7 | 5 |
| average density (2 mpf) | 44 | 29 | 24.6 | 26.8 | 22.8 | 17.7 | 14.3 | 19.5 | 18.1 | 20 | 20.7 | 20 |
| matures/box | 1.67 | 8.93 | 16.17 | 16.17 | 7.19 | 12.94 | 9.93 | 12.81 | 11.56 | 15.14 | 11.14 | 16.4 |
| matures/total (%) | 3.8 | 30.8 | 65.8 | 60.4 | 31.5 | 73 | 69.6 | 65.8 | 63.8 | 75.7 | 53.8 | 82 |
| mortality (%) | N/A | N/A | 8.1 | 8.9 | 47.7 | 20.2 | 28.1 | 28.4 | 34 | 24.3 | 42.1 | 13 |
| (matures + dead)/total (%) | N/A | N/A | 74.0 | 69.4 | 79.3 | 93.3 | 97.7 | 94.2 | 97.9 | 100 | 95.9 | 95 |
| sex ratio. % females | N/A | 41.6 | 51.7 | 46.7 | 55.1 | 52.1 | 46.4 | 48.8 | 43.1 * | 48.5 | 46.5 | 51.2 |
| sex ratio % fem. (wks 8–13) | N/A | N/A | 35.5 ** | N/A | N/A | 42.2 * | 41.4 * | 28.9 ** | 25.3 ** | N/A | N/A | N/A |
| sex ratio % fem. (wks 14–18) | N/A | N/A | 59.7 ** | N/A | N/A | 53.1 | 53.6 | 60.5 ** | 57.5 | N/A | N/A | N/A |
| median mature age (wks) | N/A | N/A | 15.8 | 16.8 | 16.6 | 14.2 | 12.6 | 14.0 | 13.2 | 13.9 | 13.9 | 12.6 |
| age of next gen. parents (wks) | N/A | 15.9 | 12.4 | 14.3 | 14.6 | 11.6 | 11.4 | 11.7 | 12.6 | N/A | N/A | N/A |

"Traditional" denotes the application of the traditional culture method (7) to the original polymorphic strain. FF(1–10)G represents the generations of selected worms. "poly" refers to the application of the final retained protocol to the polymorphic strain. "poly_4s" is the same polymorphic control but with a further selection of 10 dpf displaying four segments, similar to the FF strain. The food regimes are as follows: (A) Traditional regime - 1st month with 3ml of microalgae twice per week, 2nd month with 3ml of Sera micron twice per week, 3rd month with mixed Tetramin twice per week. (B) Regime B - 1st month with 3ml of microalgae twice per week, 2nd month with 1ml of Sera micron twice per week, 3rd month with mixed Tetramin/spinach as shown in Table 1. (C) Regime C—Sera micron used for two weeks and spinach once per week starting from the 2nd month, as described in Table 1. Chi-squared tests: * different from parity with 0.01 > p > 0.001, ** different from parity with p < 0.001.

During the selection process, we encountered two issues that significantly impacted the viability and maturation age of the worms, leading us to gradually modify our approach. The first issue was that some of the worms in certain boxes were unable to spin tubes. These worms would wander for the first two months and eventually assume a curled position with reduced mobility and feeding activity. This problem culminated in the FF4G generation, resulting in high mortality and a decrease in maturation (Fig 2). Changing the water did not correct this syndrome, and since it was not due to water fouling, we hypothesized that it might be caused by nutritional deficiency. As a result, we added ground spinach for the last four generations, and the abnormal behavior decreased from the 5th generation onwards. We then decided to provide spinach from the beginning of the second month along with Sera micron, as we reasoned that spinach must contain vitamins or other essential nutrients that are not present in Sera micron. This new approach eliminated the syndrome.

The second problem was mortality due to water fouling, which mostly resulted from feeding the worms with Sera micron. Initially, we provided quantities of the suspension that were too large for the small number of worms (15–30) in each box. However, Sera micron, which is primarily made with *Spirulina* cyanobacteria, is a highly nutritious food and efficient for fast growth of the worms. Instead of eliminating it completely, we provided smaller quantities of the food suspension and limited the period of Sera micron to two weeks, as the worms quickly gained weight and could handle the larger particles of Tetramin. Initially, we attempted to

raise the worms without performing time-consuming water changes to simplify the culture process. We believed that with a small number of worms, a controlled feeding regime, and the continued presence of dense populations of ciliates and flagellates in the boxes to clear unconsumed food, water fouling should be minimized. However, we reintroduced water changes at 60 days post-fertilization (dpf) to prevent water fouling in a few boxes that could have resulted in mass worm death. Water fouling typically results in yellowish and cloudy water. In contrast, the accumulated faeces do not seem to cause any problems for the worms despite the dirty aspect of old boxes. The box bottoms are also typically covered with green filamentous algae inoculated from the previous generation of boxes, which the worms occasionally feed on without issue (see commensals section).

The last change we conducted related to food was replacing lab grown fresh *Tetraselmis marina* microalgae with commercially frozen microalgae of the same species (see Materials and Methods). We calculated a dilution factor for the microalgae that was equivalent to the dose of fresh algae we had been using. Half of the FF8G boxes were fed fresh algae, and the other half were fed frozen algae. We took care to compare boxes from the same parents. After 6 weeks of growth, we did not observe any significant differences in growth between the two types of food (Fig 5B). Therefore, the time-consuming task of culturing the microalgae can be replaced with relatively inexpensive commercial algae, given the small quantities used.

## Worm density

The traditional method for raising worms involves alternating between high-density and low-density boxes [34]. Batches of small juvenile worms obtained from a single pair of worms in beakers are transplanted into one to three growth boxes, depending on the perceived density of worms. These worms will produce a dense layer of small tubes at the bottom of the box after a few weeks (S2B Fig). However, their growth is slow, and maturation will occur very slowly if the density remains high (our observations) [34, 37]. Increasing food doses does not help because small worms seem to inhibit each other's growth. Therefore, high-density boxes are transplanted into low-density boxes. The recommended density of worms for fast growth has typically been 30 worms per square box (around 500 individuals.m$^{-2}$) [34]. Consequently, we decided to transplant low-density boxes directly from beakers containing 10 dpf feeding and growing little worms. Once food and water fouling problems were resolved, we observed that the overall survival of these small, selected juveniles up to sub-adult age (2 mpf) was very good, and their growth was quick (Fig 4, FF(6–8)G). To determine the optimal worm density for survival, growth, and maturation, we transplanted a subset of 14 boxes of the FF10G generation with different numbers of small worms (Fig 5A). Two factors play in opposite directions: less worms in a box will grow and mature more quickly; on the other hand, we want our boxes to produce more matures over a given period of time. Growth and survival of transplanted juvenile worms is good up to 25 worms per box (Fig 5A, green curve). 25 worms/box is also the density that gives the highest productivity of mature worms after 130 days (boxes are then almost empty, Fig 5A, blue and red curves). Based on these data, we determined that the optimal density was 25 worms per box (375 individuals.m$^{-2}$), significantly less than the traditional conditions.

## Biological data comparisons

To understand how the worms in the FF culture manage to mature much earlier than the worms in our traditional culture, we compared the weights and sizes of worms from both cultures (Fig 5C, 5D, Table 3). The weights of mature worms in the FF culture are less than half those in the traditional culture. FF matures also have 10 (males) to 15 (females) segments less

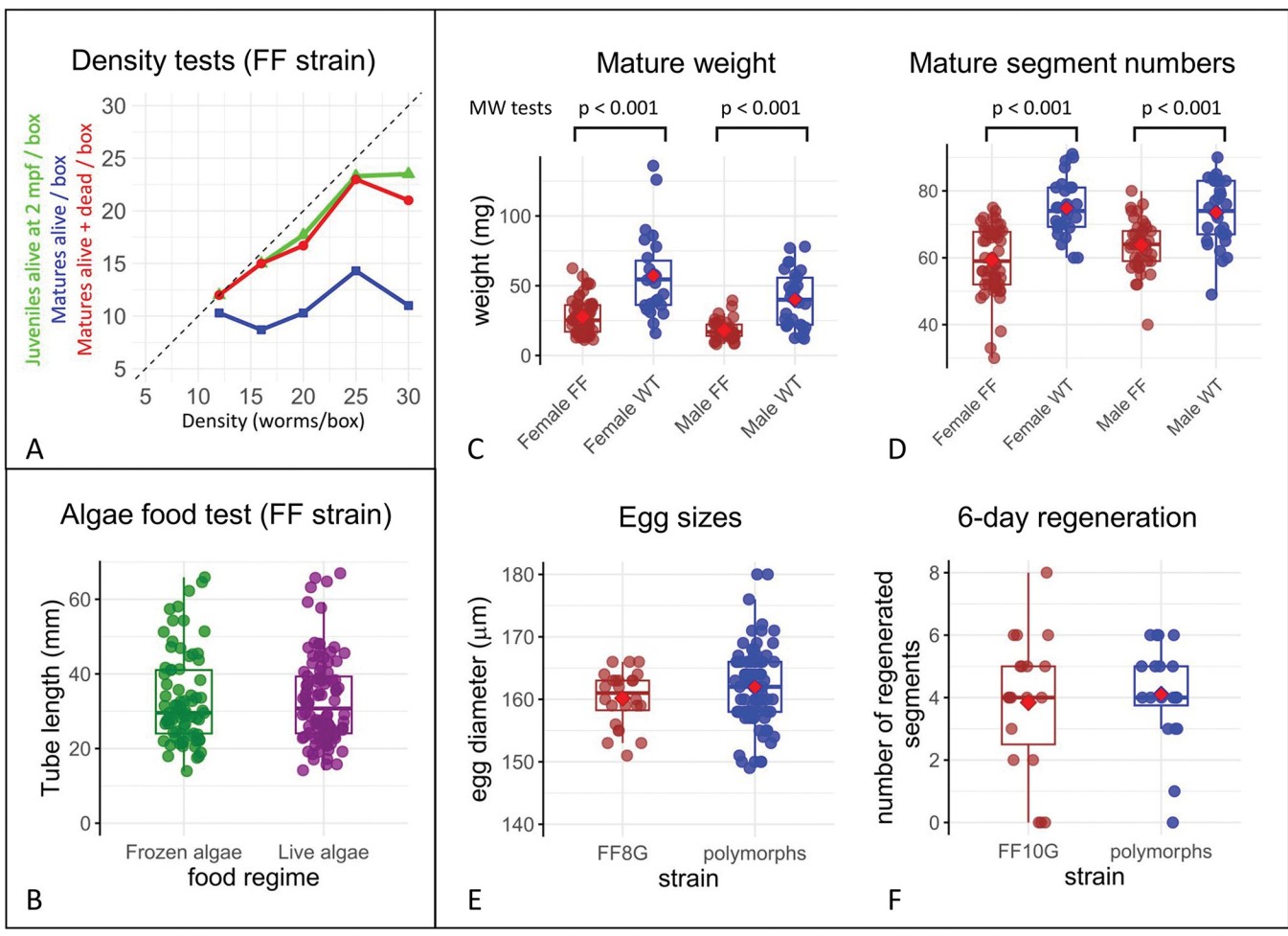

**Fig 5. Plots of biological controls and tests on the FF strain.** (A) Graph showing the effects of the density of transplanted young worms per box (at 10 dpf) on survival (green curve) and maturation. The maturation data is presented considering only the matures collected alive (blue curve) or including the dead matures found after weekends (red curve). (B) Comparison of the growth rate of FF culture worms when fed frozen algae (green) versus fresh algae (purple). To estimate growth, the length of tubes, covering the bottom of the boxes, and closely corresponding to worm length [21] was measured after two months using box photographs and ImageJ. The two populations do not conform to a normal distribution (Shapiro test). A Mann-Whitney test rejects their dissimilarity (p = 0.99). (C) Comparison of the weights of mature individuals between the FF and control (traditional culture) strains. Mature individuals were extracted from tubes before the release of gametes, gently wiped of excess water with a paper towel, and weighed on a precision balance. Each sex was compared separately as males are slimmer than females. The populations are not normally distributed (Shapiro test). The Mann-Whitney test rejects the similarity of FF and control weights for each sex (P < 0.001). (D) Comparison of the number of segments in mature individuals between the FF strain and control strain. Mature males and females have a fixed number of "thoracic" segments (15 and 22 respectively) and a highly variable number of "abdominal" segments. Animals were reversibly immobilized in a mix of 50% seawater-50% $MgCl_2$ 7.5% for 20 minutes to count segments. The populations are not normally distributed (Shapiro test). The Chi-square goodness of fit test rejects the similarity of FF and control segment numbers for each sex (p < 0.001). (E) Comparison of egg size between the FF and control strain. Several batches of eggs were sampled, and the diameter of the eggs was measured immediately after fertilization using micrographs and ImageJ. Specifically, only the largest diameter at the equatorial level, orthogonal to the animal-vegetal pole, was considered. The two populations appeared to be normally distributed, and a t-test was performed (p = 0.19) rejecting dissimilarity. (F) Regeneration capacity of the FF strain. FF and control strains were assessed by amputating a group of polymorphic and FF worms at the half-body level and allowing them to regenerate for 6 days. The number of new segment anlagen observed at the end of the 6-day period was used as an indicator of the worms' regeneration capacity and speed. The two populations fit a Poisson distribution better and were compared with a Chi-square goodness of fit test (p = 0.68), rejecting dissimilarity.

than the polymorphic matures. Early maturation with the new culturing method clearly results in smaller mature worms. We then asked whether selected smaller females would have fewer eggs or possibly smaller eggs. We measured egg diameters and found that they were indistinguishable, at around 160–165 μm, in FF-raised females and traditionally raised females (Fig 5E, Table 3). Since the mass of a mature female's body is mostly composed of oocytes, this implies that early maturing

**Table 3. Comparative tables between FF and polymorphic strains.**

|  | "Fast Forward" protocol | "Traditional" protocol |
|---|---|---|
| Temperature | 20˚C | 18˚C |
| Day / night (hours) | 16 / 8 | 16 / 8 |
| Moonlight | not necessary | required for sexual metamorphosis synchronization |
| Density (worms / m$^2$) | 375 | alternance of high density (>2000) and low density (500) |
| Food | Microalgae, Sera micron, spinach, Tetramin in finely ajusted quantities | Microalgae, Sera micron, spinach, Tetramin |
| Selection of healthy juveniles | required | not required |
| Change of water | every 2 months | every two weeks |
| Bubbling | not required | required for high density |
| Average mature weight (mg) | 27.7 (female) 18.1 (male) | 57.2 (female) 40.1 (male) |
| Number of segments | 59.3 (female) 63.8 (male) | 74.8 (female) 73.5 (male) |
| Average mature age (weeks) | 13.2 | >18 |
| Egg size (μm) | 161 | 162 |

FF females will have roughly half the number of oocytes compared to traditionally raised females. Nevertheless, one FF female typically releases more than 1000 oocytes, which is more than enough for all the intended uses in the lab (micro-injections, colony reproduction, immunohistochemical or in situ hybridization labelings). We also wondered whether the regeneration capabilities of the selected FF juveniles were affected. A comparison of regeneration speeds between FF and traditionally raised polymorphs revealed no difference in this regard (Fig 5F, Table 3). Last and most importantly for a model that is used for reverse genetics experiments, the eggs have been micro-injected with success numerous times in our team.

## Discussion

As described above, the life cycle of *P. dumerilii* is lengthy and complex. This has hindered the widespread adoption of this model on a global scale. In response to this challenge, we propose in this article a new and fast-reproducing strain called "Fast-Forward" (FF). The "Three Rs" principle outlined in Directive 2010/63/EU of the European Union legislation reflects the scientific community's current trend to Replace, Reduce, and Refine animal experimentation. The protocol presented in this article aims to reduce the resources required to culture *P. dumerilii* while accelerating and simplifying the management of its culture conditions. According to the Refinement principle of the "Three Rs", we anticipate that the proposed modifications to the care practices, notably food quantity and density, will alleviate the distress experienced by the animal, thus reducing the variability of scientific results. Improving data quality indirectly contributes to the Reduction principle, as fewer animals are required to obtain valuable results. However, the use of live animals remains a widely used strategy in developmental genetics, and it is an area that must be improved in the future.

### Suitability of the fast culture for different types of biological studies

The original laboratory culture of *P. dumerilii* was developed in 1953 in Germany by Carl Hauenshild and Albrecht Fischer [7, 19]. Since then, all laboratories that have developed

cultures appear to have used offshoots of this original German culture. This strain initially included a mixture of Atlantic and Mediterranean worms, which may partly explain the high level of genetic polymorphism. A complete culture method was posted online for a long time on the Platynereis.de website, which was cited in many previous articles and helped many groups establish their own culture. Although this website is no longer available, a copy of the method can be found on the Platynereis.com website. Several groups have since attempted to simplify the method. A recent effort has resulted in a "scalable culturing system" [34] for starting a small-scale culture without a dedicated thermostatic room. This method simplifies the food regime, alternating between *Spirulina* powder for the youngest worms and Sera micron for juveniles up to sexual maturation.

The FF protocol offers several steps for simplification, enabling the compaction of culture and preservation of wild-type stock and several different genetically modified strains in a limited space. We use flat boxes that can be stacked with at least two boxes on each shelf, without hindering gas exchange into the water. Piling up to four boxes has also been done without any issues. However, it is important to use plastic boxes with non-airtight lids (refer to Materials and Methods). We do not use air bubbling at any stage, due to the low-density of the culture and carefully managed food regimen. Nevertheless, we have effectively maintained high-density boxes using the same food regimen, eliminating the necessity for air bubbling. These high-density boxes remain of interest, either as backup for transgenic strains (as discussed below) or for the transplantation of time-shifted low-density boxes (as described below). Compared to earlier methods, this protocol significantly reduces the need for water changes. With only one water change required when the worms reach two months (instead of every two weeks), a low-density box of worms can complete its cycle using only one litre of NFSW for a maximum of four months. This minimizes the workload, expenses, and natural resources required, making it possible to maintain bigger cultures in the future. Lastly, our protocol eliminates the need for using an artificial moon to synchronize the spawning periods of males and females. Importantly, moonlight, whether natural or artificial, has been shown to play a crucial role in synchronizing a pre-existing endogenous circalunar clock [26]. However, it does not, by itself, trigger sexual metamorphosis. The absence of moonlight therefore does not delay maturation if all other conditions are good. The maturation and swarming of FF-cultured worms occur over such a short time span that no synchronization of sexual cycles is necessary to obtain males and females on a daily basis. We continued following this culture protocol until the tenth generation (FF10G), and we did not observe any degradation in the health or reproduction efficiency. The only clear drawback to this efficient maturation process is the necessity to collect adults every morning on weekdays to prevent their death in the boxes and to clean off any dead adults (usually one or two per box) after a weekend.

The main advantage of this culture technique is the acceleration of the reproduction cycle, which is a crucial step in establishing *P. dumerilii* as a valuable model for transgenesis and genome editing techniques. Measuring the average life cycle of *P. dumerilii* has seldom been done in the past, but previous methods have clearly resulted in much longer life spans and, consequently, reproduction cycles. The culture technique used at the Institut Jacques Monod for previous works since 2009, following the Hauenschild/Fischer method, involved using high-density boxes for an extended period (3–4 months) before worms were transferred to low-density boxes for maturation. As a result, very few worms matured before 4 months, and even after dispatching to low-density boxes, few worms matured before 5–6 months. The recently proposed scaled-down, cost-effective in-lab culture [34] does not focus on speeding up the cycle and maintains high-density boxes (more than 300 worms/box) for two months after fertilization. A work on *corazonin* effects on growth and maturation [37] mentions a median maturation time for control polymorphic worms of around 8 months, compared with

3.2 months with the FF rearing protocol. In contrast to these lengthy periods of culturing at high density, the crucial parameters manipulated in the current technique are the immediate establishment of low-density boxes and the careful selection of fast-growing small worms at ten days.

This culturing technique may not suit all experimental objectives. For instance, research teams investigating biological rhythms, particularly those influenced by the lunar cycle [38], find limitations in our approach due to the complete absence of an artificial moon cycle, preventing these specific studies. Moreover, the swift maturation of most worms within a single moon cycle (28 days) proves inadequate for such inquiries, even if a moonlight cycle is reintroduced in the incubator. Other studies focusing on time-related processes such as regeneration [39] rely on meticulously controlled culture conditions with recorded timelines. When utilizing the FF strain and its fast-cycling protocol, it becomes imperative to evaluate the selected worms' capabilities in the specific process under study and subsequently establish new standard timelines. However, our observations indicate no significant differences in this process when comparing traditional and FF strains. The discontinuation of high-density boxes in this culturing method hampers the maintenance of substantial quantities of medium-sized juveniles, presenting practical challenges for teams exploring phenomena occurring at these developmental stages. Nevertheless, as elaborated below, high-density boxes can still be upheld under the same conditions, necessitating a water change every two months.

Interestingly, worms of our polymorphic culture raised in the same conditions as the FF strain grow and mature as rapidly as the FF strain when considering the median age of maturation. A fast culture can be started by any lab that already possesses the polymorphic strain. This does not mean that the artificial selection we have carried out has been ineffective. We have tested only a limited number of biological characteristics and only the size and weight make a difference between worms of the FF and traditional culture protocols. Some other questions may be answered in the near future. For instance, can the FF strain worms be kept in high density cultures for as long a period of time as the polymorphic strain?

## Comparison with other metazoan models

Table 4 provides an overview of current metazoan models that have been acclimated in laboratory settings for experiments involving genetics and reverse genetics. This list includes a mix of models that have been developed for a long time, some for more than a century, with well-established protocols and numerous publications utilizing these techniques (such as the house mouse, fruit fly, and *Caenorhabditis elegans*), as well as several other models where efforts to develop genetic approaches are still in their early stages. Only a few models are available for studying the vast diversity of genetically dependent biological phenomena, as the duration of the life cycle is often a significant obstacle.

The current list of short evolutionary branch metazoans includes notable species such as amphioxus [71], hemichordate worms [72], *P. dumerilii* [6], the myriapod *Strigamia maritima* [73], and the sea anemone *Nematostella vectensis* [74], representing chordates, deuterostomes, annelids, arthropods, and cnidarians, respectively. All these species occupy key phylogenetic positions for reconstructing ancestral states in the metazoan tree. Phylogeny and evolution are not the only motivations for developing new genetic models, as some species exhibit unique derived characters that are worth exploring at the genomic/genetic level, such as the tardigrade *Hypsibius exemplaris* [75] and its ability to undergo anhydrobiosis.

Our new protocol places *P. dumerilii* in a better position for development as a high-performing genetic model in the future. The median life cycle obtained with this method (around 14 weeks) places *P. dumerilii* in a comparable position to the widely used fish model *Danio*

**Table 4. Reproductive characteristics of animal laboratory models used for transgenesis/genome editing experiments.**

| Phylum | Class | Common Name | Animal Model | Generation Time | Number of Offsprings | References |
|---|---|---|---|---|---|---|
| Annelida | Clitellata | Earthworm | *Lumbricus terrestris* | 6–12 months | 3.7/month | [40] |
| Annelida | Clitellata | Earthworm | *Eisenia foetida, Eisenia andrei* | 2–3 months | 1-20/spawning | [10, 41] |
| Annelida | Clitellata | Leech | *Helobdella robusta* | ~ 2 months | 267/spawning | [11] |
| Annelida | Clitellata | Leech | *Helobdella triserialis* | 30–35 day | 302/spawning | [42] |
| Annelida | Clitellata | Leech | *Helobdella octatestisaca* | 140 days | 119/spawning | [42] |
| Annelida | "Polychaete" | Ragworm | *Platynereis dumerilii* (polymorphic stock) | 6–8 months | up to 2000, semelparous | [18, 43] |
| Annelida | "Polychaete" | Ragworm | *Platynereis dumerilii* (FF strain) | 3.5 months | more than 1000, semelparous | this work |
| Annelida | "Polychaete" | Sandworm | *Capitella teleta* | 8–10 weeks | 100-250/brood | [44] |
| Mollusca | Cephalopoda | Octopus | *Octopus vulgaris* | 8–9 months | tens of thousands/spawning | [45] |
| Arthropoda | Branchiopoda | Water flea | *Daphnia magna* | 7–10 days | hundreds/lifetime, parthenogenesis | [46] |
| Arthropoda | Insecta | Fruitfly | *Drosophila melanogaster* | 10 days | hundreds/spawning | [47] |
| Arthropoda | Insecta | Red Flour Beetle | *Tribolium castaneum* | 4 weeks | 2-5/day | [48, 49] |
| Nematoda | Chromadorea | Roundworm | *Caenorhabditis elegans* | 3.5 days | ~300/spawning | [50] |
| Chordata | Actinopterygii | Zebrafish | *Danio rerio* | 60 days | 1-700/spawning | [51] |
| Chordata | Actinopterygii | Medaka, ricefish | *Oryzias latipes* | 3–4 months | daily spawning | [52] |
| Chordata | Actinopterygii | Cavefish | *Astyanax mexicanus* | 6–8 months | hundreds/spawning | [53] |
| Chordata | Amphibia | African clawed frog | *Xenopus laevis* | 1 year | 400-1000/spawning | [54–56] |
| Chordata | Amphibia | Frog | *Xenopus tropicalis* | 4–5 months | 1.000–3.000/spawning | [55] |
| Chordata | Amphibia | Salamander | *Ambystoma mexicanum* | 1 year | 200-600/spawning | [57] |
| Chordata | Mammalia | House mouse | *Mus musculus* | 9–11 weeks | 4–9/litter | [58, 59] |
| Chordata | Ascidiacea | Ascidian | *Ciona intestinalis* | 1–2 months | 2000-3000/spawning | [60] |
| Chordata | Actinopterygii | Stickleback | *Gasterosteus aculeatus* | 6 months | 40-450/spawning | [61] |
| Echinodermata | Echinoidea | Sea urchin | *Temnopleurus reevesii* | 3–4 months | hundreds of thousands/spawning | [62] |
| Echinodermata | Echinoidea | Sea urchin | *Strongylocentrotus purpuratus* | 11 months | hundreds of thousands/spawning | [63] |
| Cnidaria | Hexacorallia | Sea anemone | *Nematostella vectensis* | 8–10 weeks | tens-hundreds/spawning | [64] |
| Cnidaria | Hydrozoa | Green Hydra | *Hydra viridissima* | 6.6 ± 1.5 days | 1 /spawning | [65, 66] |
| Cnidaria | Hydrozoa | Moss polyps | *Hydractinia echinata, H. symbiolongicarpus* | 3 months | daily spawning | [67] |
| Cnidaria | Hydrozoa | Jellyfish | *Clytia hemisphaerica* | 2 months | hundreds/spawning | [68] |

*D. melanogaster* and *C. elegans* are exceptions in this regard, as they depend on ephemeral food resources in the wild, specifically rotting fruits. As a result of selection pressure, they have evolved exceptionally fast development, growth, and sex maturation processes. However, these widely used models with fast life cycles are often correlated with rapid evolution at the molecular and developmental levels, making them "long evolutionary branch" models [48, 69, 70]. Therefore, there is great interest in developing alternative model organisms with a short evolutionary branch for genetic investigations. These models can provide insights into the emergence of major anatomical features, cellular processes, and genetic machineries, simply because they are less derived in these respects compared to well established models.

*rerio* (zebrafish, Table 2). Among annelids, the fast culturing of P. *dumerilii* places it on par in terms of culture conditions with other models, including both clitellates and "polychaetes" that have been developed in the past. While clitellates have interesting developmental features such as large size and easily micro-injectable teloblasts, they are derived annelids, adapted to freshwater or terrestrial lifestyles. Among "polychaetes", *Capitella teleta* [44] has emerged with many publications in the last two decades. It belongs to the large clade Sedentaria. Lineage tracing by micro-injection [76] and genome editing by CRISPR-Cas9 [77] have been used successfully in this species. More recently, the other sedentarian *Owenia fusiformis* [78, 79] has also been shown to possess valuable characteristics for diverse biological studies. *P. dumerilii* however displays two main advantages over these models: being an errantia of the family

Nereididae, it is widely considered to possess a more ancestral anatomy than the sedentarians and the reproduction by epitokous swarming allows for complete control of the reproduction event and easy manipulation of thousands of eggs and embryos, obtained daily using the fast-cycling culture method described here.

## Temperature of culture

The temperature used for the Fast Forward strain is two degrees higher than the traditional temperature (18˚C) commonly used in most publications and notably in the standard chart of *Platynereis* development [18]. *Platynereis* embryonic and larval development are highly influenced by temperature. We noticed that juveniles reared at 20˚C began feeding at 4–5 days instead of the usual 6 days. Incidentally, observations hinted that worms raised on our lab benches at room temperature (ranging from 21 to 23˚C in summer) grew notably faster than those in the controlled environment at the traditional 18˚C. Consequently, we opted to set our incubators at the highest temperature feasible (20˚C). We maintain a separate small incubator for batches of eggs requiring development at the standard temperature (18˚C). Research groups using only a controlled room temperature can adhere to the standard temperature. In such cases, it is likely that the median maturation age might slightly increase, although we did not conduct specific tests on this aspect.

## Culture size

The packing of boxes in incubators allows for easy scalability without the need to dedicate a thermostatic room of adequate size and shape for culture. The elimination of the lunar cycle and the rapid maturation of worms after 10 weeks are also important factors. There is no need for two rooms with alternating lunar cycles to obtain mature individuals over a monthly period. The maturation peaks will only depend on the age of the boxes. Mature worms will be used for three main purposes: colony maintenance, crosses with transgenic/CRISPR-edited worms, and micro-injection. We have generally been selecting mature worms aged less than 90 days to start the next generation of the FF population, and this selection is currently maintained. Worms older than 90 days are used for genetics and micro-injection. However, *P. dumerilii* still displays some unfavourable characteristics for genetic experiments, such as having 2n = 28 chromosomes, which limits the possibility of complex genetic combinations. Maintaining homozygous strains will be challenging as it requires two homozygous adults of opposite sexes on the same day, which in turn requires a large pool of worms for each strain. Therefore, the solution for keeping homozygous strains may lie in the development of sperm (or even embryo) freezing techniques [80]. Transgenic lines expressing fluorescence as heterozygotes are maintained by crossing transgenic worms with wild-type FF individuals and selecting transgenic progeny under an epifluorescence microscope as early as 10 days post-fertilization. Our group in Paris has successfully maintained three different strains in this manner with only four boxes, each containing 20 worms, for each strain.

In the current culture conditions, assuming an initial seeding of 20 worms per box and a conservative mature collection efficiency of 60%, we can calculate a productivity of approximately 1–1.2 worms/box/week over a 10-week period. To counter the small difference in the temporal maturation of males and females, it is advisable to use batches of eggs that are obtained over a period of two weeks. Early males of the younger boxes can be mated with females of the older boxes. There will be an unproductive period of 8–10 weeks following, as a new generation of boxes is growing. For teams that require wild-type worms throughout the year, which will be the case if multiple transgenic strains need to be maintained, the number of boxes will need to be doubled with a shift of 8–10 weeks between the seeding of low-density

boxes. This shift can be easily achieved by maintaining batches of the same generation of small worms in high-density boxes (>300 worms) from 10 dpf on, fed exclusively with algae, for 8–10 weeks. The worms will remain very small until they are used to create new low-density boxes.

The current study may not represent the ultimate effort in establishing the most efficient culturing system for *P. dumerilii*, but it is a significant step towards reducing the variability in the age of sexual maturation. We believe that there is still much progress to be made in improving the food regime, particularly during the early settlement of the worms. One potential solution could be to draw inspiration from the recent small-scale culture protocol [34], where they reduced the food regime to two items, as opposed to the four items utilized in our present method. Another area for improvement could be the replacement of natural filtered seawater, which can be costly for facilities not located near the sea, with artificial seawater. Additionally, using natural seawater implies that its quality may vary through the year. Environmental factors can interfere with the reproductive cycle of the Nereididae family: endocrine control and chemical communication between individuals have a major role in regulating Nereididae's development and reproduction [81]. The effects of water seasonality on worms' maturation have not being investigated yet, but, for the reasons listed above, they are expected to have minor contributions. Hopefully, our findings, combined with advancements in developing efficient transgenesis, will encourage more groups in the future to adopt this remarkable model animal for their own studies and establish their own culture of *P. dumerilii*.

## Supporting information

**S1 Fig. Sex ratio in mature animals across the selected FF generation.** FF(2–8)G are the selected generation for which significant numbers of matures have been obtained. "poly" is the control culture with polymorphic worms raised with the same density and food regime as the last three selected FF generations.
(TIF)

**S2 Fig. Worm boxes and incubator.** A. An open incubator is shown, displaying the stacking of worm boxes on shelves. Typically, boxes are stacked in pairs, allowing for the placement of 8 boxes on each shelf (half of the boxes are hidden in this photograph), totalling 50 boxes in the incubator. Take note of the LED ribbon dispensing warm white light, which is affixed all around the frame of the front door, approximately 2 cm from the edge, and connected to a mechanical timer (located at the top of the incubator). The box lids are secured in two corners, leaving the other two corners unfastened to facilitate gas exchange. A convenient method for feeding the boxes involves partially pulling out the shelves to access the inner boxes, and the unclipped corners are raised a few centimetres to allow the dispensing of food using a plastic pipette. B. This image shows a 2-month-old high-density box, illustrating the arrangement of the worm tubes, which are spaced apart from each other. C. In this photograph, a 2-month-old low-density box with 25 worms is depicted, highlighting the faster growth of the worms (notice the significantly longer tubes), which enables maturation to commence two weeks later.
(TIF)

**S1 File. Troubleshooting table.**
(DOCX)

**S2 File. Equipment and consumables table.**
(DOCX)

## Acknowledgments

We thank M. Kapsimali for her help in R scripting and L. Guzzetti for help in statistical analysis. We thank P. Kerner for his help in microalgal supplies and tips on using frozen algae. We acknowledge the staff of the animal facility of the institute J. Monod for help in worm husbandry.

## Author Contributions

**Conceptualization:** Guillaume Balavoine.

**Data curation:** Mathieu Legras, Giulia Ghisleni, Léna Regnard, Manon Dias, Rabouant Soilihi, Enzo Celmar, Guillaume Balavoine.

**Formal analysis:** Mathieu Legras, Giulia Ghisleni, Guillaume Balavoine.

**Funding acquisition:** Guillaume Balavoine.

**Investigation:** Mathieu Legras, Giulia Ghisleni, Léna Regnard, Manon Dias, Rabouant Soilihi, Enzo Celmar, Guillaume Balavoine.

**Methodology:** Mathieu Legras, Giulia Ghisleni, Léna Regnard, Manon Dias, Rabouant Soilihi, Enzo Celmar, Guillaume Balavoine.

**Project administration:** Guillaume Balavoine.

**Resources:** Giulia Ghisleni, Guillaume Balavoine.

**Software:** Giulia Ghisleni, Guillaume Balavoine.

**Supervision:** Guillaume Balavoine.

**Validation:** Giulia Ghisleni, Guillaume Balavoine.

**Visualization:** Mathieu Legras, Giulia Ghisleni, Guillaume Balavoine.

**Writing – original draft:** Mathieu Legras, Giulia Ghisleni, Guillaume Balavoine.

**Writing – review & editing:** Mathieu Legras, Giulia Ghisleni, Guillaume Balavoine.

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
