## [Decision Letter · Decision Letter 0]

30 Oct 2023

PONE-D-23-30265Fast cycling culture of the annelid model Platynereis dumeriliiPLOS ONE

Dear Dr. Balavoine,

Thank you for submitting your manuscript to PLOS ONE. After careful consideration, we feel that it has merit but does not fully meet PLOS ONE’s publication criteria as it currently stands. Therefore, we invite you to submit a revised version of the manuscript that addresses the points raised during the review process.

As requested by the two reviewers, the manuscript needs to be corrected in some points in order to increase the clarity of the experiments and results. Therefore, I ask you to try to modify the text following the advice of both reviewers.

We look forward to receiving your revised manuscript.

Kind regards,

Hector Escriva, PhD

Academic Editor

PLOS ONE

Journal Requirements:

"The Balavoine group was financially supported by the CNRS, the University Paris Cité, the Institut Jacques Monod, the Agence Nationale de la Recherche (grant PRCI TELOBLAST ANR-16-CE91-0007) and the Fondation ARC pour la recherche sur le cancer (grant LSP 190375). GG was supported by a Master student fellowship of the EUR G.E.N.E. graduate school (#ANR-17-EURE-0013) that is part of the Université Paris Cité IdEx #ANR-18-IDEX-0001 funded by the French Government through its “Investments for the Future” program."

5. Please expand the acronym “ARC” (as indicated in your financial disclosure) so that it states the name of your funders in full.

Reviewers' comments:

Reviewer's Responses to Questions

**Comments to the Author**

1. Is the manuscript technically sound, and do the data support the conclusions?

Reviewer #1: Partly

Reviewer #2: Yes

2. Has the statistical analysis been performed appropriately and rigorously? 

Reviewer #1: I Don't Know

Reviewer #2: Yes

3. Have the authors made all data underlying the findings in their manuscript fully available?

Reviewer #1: No

Reviewer #2: Yes

4. Is the manuscript presented in an intelligible fashion and written in standard English?

Reviewer #1: Yes

Reviewer #2: Yes

5. Review Comments to the Author

Reviewer #1: The manuscript describes a new faster culturing method for the marine model system Platynereis that will improve its applicability for the scientific community. Using a small number of worms per box and different food is the main conclusion of the work. A strength of the work is the clear and nice visually aid of the life cycle, batch-making, and commensals.

Unfortunately, the text is not always clear and some phrases lead to confusion. Some conclusions are not supported by references or their own data. Adding additional analysis of data, e.g., biological replica (=boxed), or data from the traditional culture (if existing) would further strengthen the work. Statistical analyses could not be checked, because the raw data is not public yet.

I recommend the manuscript with minor revision. After revising, I believe the manuscript will be a great asset.

Major comments:

1) Raw data, including data from single boxes and different experiments (e.g., frozen vs fresh algae) have to be made public when published

2) The authors did not show clearly that worms from the new culture reach maturation earlier than under classical conditions. Could the authors include data and a plot in Fig 4 for the traditional culture? I believe this data point is crucial to show the improvement of the maturation time with the new method, and supports the conclusion that the new culture results in earlier maturation. The information would also strongly support that the animal density in the boxes is a crucial factor.

3) The description of the different experimental conditions was not always clear to me. Using ‘traditional’ or ‘classical’ food and ‘traditional’ or ‘classical’ method in the text is sometimes ambiguous. Often it was not clear if the authors meant their own old lab culture or Hauenschild & Fischer protocol, or/ and ‘classical’ food from table 2? Does classical method refer to food, temperature (18C or 20C) or worms per box?

Suggestion: A table with the experiment-name and all conditions might be a useful way to summarize, e.g., temperature, food regime, artificial selection of maturation time and 10dpf larvae; for ‘poly’ and ‘poly4’ generation: which conditions were introduced (food, low density box).

4) Concentration of MgCl2, antibiotics and food are not clear in the text

5) Correction of the following wording is required:

- L 91: ciliary belt TO ciliary band

- L 161: swimming adults TO matured worm or epitoks

- silk tube TO mucous tube

- genetic selection TO artificial selection

6) L414: The authors describe that the number of worms per box in FF10G is different between the boxes. It is not clear to me, how it is reflected in table 2 and fig 4, and accounted for?

Therefore, the variance within a single generation (FF, poly, and traditional) is an important factor and should be included in the manuscript.

7) I would like to ask the authors to explain their following conclusion:

- L517: There has been no difference between the poly and poly4 generation observed, if I understood correctly. Why is the selection of 10dpf old larvae important?

- L411-413: 'Mortality rate is nearly non-existent.' I was not able to find the corresponding data

- L 376: Missing spinach in diet suggest deficit. To my knowledge, Kuehn et al. 2019 did not report problems in diet without spinach. I was wondering if there might be other factors?

- L 385: It was stated, that water change was reintroduced in a few boxes – How did the authors control for this, and how is this reflected in the data?

8) The authors included some unsupported statements and speculations. I would like to ask the authors to excluded those from the text or support them with their data or a reference. I only mention some instances.

- L 413: “growth was quick”

- L556..: paragraph about evolutionary branches

- L 400: “Worm density … most important…”

- L405: “…maturation will not occur if density remains above 100 worms”

- L78-79: function of jelly in Fisher & Dorresteijn 2004 is described differently and I was not able to find citation showing decreased predation due to jelly

- L 378: Spirulina is very efficient for fast growing worms

9) The reported maturation age in the text is inconsistent (L26 vs L 588 vs 134-135), and the maturation age partially refers to earliest or median maturation in the wrong context L348-351

Suggestion: Because the food regime changed between the FF generations, I believe it is important to point out early in the results that the shorter maturation time is caused by changed feed regime, and not artificial selection.

10) Missing references: (only examples)

- L599 reference not included in reference list

- L357 missing reference

- L69-70 transparency good for imaging and optogenetics(?)

- L 80-83: spiral cleavage

11) L29: “… not required…an artificial moonlight regime for synchronizes sexual maturation…”

If correctly understood, the artificial moon triggers synchronized spawning, but without artificial moon light, maturation is not prevented. It was not clear to me how this is different between the traditional and new culturing method. There is no data that the new culture method affects the synchronization of maturation.

12) L396 Refers to supplementary fig 2 for frozen/fresh algae, but should be Figure 5B.

13) Could the authors please clarify the following abbreviations in the text:

- L 165: NFSW

- L213/214 “w/v” and “sub-adults”

14) L 92: “The only nutrient source is the yolk contained in their macromere” I am unsure about the previous mentioned function of the lipid droplets, and I was not able to find a reference that yolk is in macromeres.

15) Could the author add the time they collected matured worms in table 2 "total matured collected? The would help to interpret especially the “classical” condition.

Minor comments:

It is up to the authors, if they consider the suggestions and questions.

1) Why does the feeding start at 4dpf, but Platynereis is only able to feet from 6days on?

2) The manuscript focuses on a new culturing method, and I believe a very detailed description of the life cycle, old non-functional website, comparison with other metazoan models, number of chromosomes and the 3 Rs might be irrelevant for this format and distract from the key message on speeding up the life cycle of the worms.

3) The culture uses natural seawater. Is there a seasonal effect of the natural sea water onto the maturation time?

4) L 178 and L 182: “…incubated … at 20C” Is there a reason the incubation happens at 20C, and not the standard 18C?

Reviewer #2: The manuscript ‘Fast cycling culture of the annelid model Platynereis dumerilii’ by Legras et al describes culture and selection conditions for an accelerated life cycle of this marine annelid, one of a few more broadly utilized spiralian model systems. They selected for a Platynereis strain FastForward FF that reaches sexual maturity at 13 to 14 weeks when fed under an improved conditions with optimal space and optimal nutrition being the main factors. This efficiency in culturing is important for the research community as it will enable the propagation of more strains in a shorter amount of time, a prerequisite to use these worms for transgenic approaches. The authors assess carefully the health of the strain, and characterize the worms at the time of sexual maturation. The work is carefully done, and well describe and documented, and another step forward to make Platynereis a useful spiralian model organism to dissect biological processes at the molecular level.

Major comments & suggestions:

1. lane 27-30 “A low worm density in boxes and a strictly controlled feeding has several advantages like not requiring air bubbling or an artificial moonlight regime for synchronized sexual maturation”. Q: How does the new culturing method impact hormonal pathways controlled by moonlight to promote maturation? Could there be long term effects for the health of a population of worms under these conditions? I see of course the advantage of a very efficient cycling culture system but what might be the trade offs? The authors should discuss this issue in more detail.

2. lane 27, 134, 355, 399 Could a concise description of the traditional protocol be included as well e.g. supplemental or direct comparison? This would make it more accessible for readers, and to gain a quicker understanding of the differences between the traditional and new culture methods.

3. Maybe it is good to create a table to compare between polymorphic worms and FF worms in the results (ex: segment, maturation, egg size, or advantage).

Minor Comments:

62 Nereididae (capitalize)

Table 1: Monday, Wednesday, Friday (capitalize)

Table 3: Addition of ascidians and sea urchins as marine invertebrates?

Figure 2 Day4 Fryday (Friday)

The scheme in Figure 2 is misleading because some arrows illustrate the flow through and others the progression of the process e.g day 2 (dejellyfication) the flow through arrow is directed to 10 dpf. Maybe remove this arrow (see red circle).

Figure 3: fairly low resolution figure

Figure 5A Density Tests or Density Test

Figure 5: A & B are outlined but C to F are not. Be consistent.

6. PLOS authors have the option to publish the peer review history of their article (what does this mean?). If published, this will include your full peer review and any attached files.

Reviewer #1: No

Reviewer #2: No

---

## [Author Response · Author response to Decision Letter 0]

16 Nov 2023

To the Academic Editor Hector Escriva, Ph.D., and to the reviewers who revised our manuscript,

We want to sincerely thank you for taking the time to review our manuscript and allow us to improve the presentation of our work. We are uploading, together with this letter, the marked-up copy of the revised manuscript and the unmarked version. We have thoughtfully considered the editor’s and reviewers’ notes and we modified and integrated the text accordingly. In this document, we answered the comments point-by-point in green color. We appreciate the referees’ positive feedback, and we believe the resolved issues have much improved the quality of the manuscript.

Guillaume Balavoine, Ph.D., and all the authors of the manuscript

Journal Requirements:

We carefully consulted the PLOS ONE templates and we adjusted the figures, tables, and supplementary information reference format. Everything should follow the journal’s guidelines.

All our data are deposited on the Dryad open data publishing platform and are available at this link: https://datadryad.org/stash/share/S0lXlKFfdfxmxyHLJUWEQvoq9vIum7bQDuX3nRGPHBw

We have added the following sentence to the methods section: “Animal research in the European Union (EU) is regulated under Directive 2010/63/EU on the protection of animals used for scientific purposes. The annelid Platynereis, as other invertebrates except cephalopods, is not covered by this regulation and no authorization is required for experimentation.” You can find this integration in the lines 170-173 of the “Revised Manuscript with Track Changes”

"The Balavoine group was financially supported by the CNRS, the University Paris Cité, the Institut Jacques Monod, the Agence Nationale de la Recherche (grant PRCI TELOBLAST ANR-16-CE91-0007) and the Fondation ARC pour la recherche sur le cancer (grant LSP 190375). GG was supported by a Master student fellowship of the EUR G.E.N.E. graduate school (#ANR-17-EURE-0013) that is part of the Université Paris Cité IdEx #ANR-18-IDEX-0001 funded by the French Government through its “Investments for the Future” program."

We have added the statement to our acknowledgments paragraph (lines 925, 926 of the “Revised Manuscript with Track Changes”).

5. Please expand the acronym “ARC” (as indicated in your financial disclosure) so that it states the name of your funders in full.

The full name is “Fondation pour la Recherche sur le Cancer”. This foundation is still designated by its old acronym ARC. The acknowledgement section has been changed accordingly. 

We revised our reference list and the quoted references are complete and correct. All the changes and additions to the reference list are highlighted in the “Revised Manuscript with Track Changes” file.

Reviewers' comments:

Reviewer #1:

Major comments:

1) Raw data, including data from single boxes and different experiments (e.g., frozen vs fresh algae) have to be made public when published

We are sorry that the reposit link provided in the manuscript file is incomplete when clicked. Here is the complete one: https://datadryad.org/stash/share/S0lXlKFfdfxmxyHLJUWEQvoq9vIum7bQDuX3nRGPHBw

2) The authors did not show clearly that worms from the new culture reach maturation earlier than under classical conditions. Could the authors include data and a plot in Fig 4 for the traditional culture? I believe this data point is crucial to show the improvement of the maturation time with the new method, and supports the conclusion that the new culture results in earlier maturation. The information would also strongly support that the animal density in the boxes is a crucial factor.

We integrated in Figure 4 the graphs for the traditional culture and 1st generation FF, for which statistics were already available in Table 2. No median maturation age has been obtained for the traditional culture. Still, the fact that only a few matures had been collected at 18 weeks is consistent with statements from the literature, as indicated in the text. 

3) The description of the different experimental conditions was not always clear to me. Using ‘traditional’ or ‘classical’ food and ‘traditional’ or ‘classical’ method in the text is sometimes ambiguous. Often it was not clear if the authors meant their own old lab culture or Hauenschild & Fischer protocol, or/ and ‘classical’ food from table 2? Does classical method refer to food, temperature (18C or 20C) or worms per box?

Suggestion: A table with the experiment-name and all conditions might be a useful way to summarize, e.g., temperature, food regime, artificial selection of maturation time and 10dpf larvae; for ‘poly’ and ‘poly4’ generation: which conditions were introduced (food, low density box).

We made sure to uniform all the protocols’ definitions throughout the manuscript. We also followed your suggestion about the experiments’ conditions and we extended Table 2 to include these pieces of information.

4) Concentration of MgCl2, antibiotics and food are not clear in the text

These imprecisions have been corrected in the text and Supplementary File 2, in which we have added complete recipes for making food suspensions.

5) Correction of the following wording is required:

- L 91: ciliary belt TO ciliary band

- L 161: swimming adults TO matured worm or epitoks

- silk tube TO mucous tube

- genetic selection TO artificial selection

We modified the text accordingly. However, we think defining the tubes as “mucous”, while being a generic habit in annelid literature, is inaccurate. In “Behavioural and Secretory Activity during Tube Construction by Platynereis dumerilii Aud & M. Edw. [Polychaeta: Nereidae]” By John M. Daly (1973), the tubes are described as follows: “The 'mucous' tube of Platynereis dumerilii is made up of successive layers of fine threads held together by an adhesive secretion”. The tube is thus made of a mix of silk (protein fibers spun in threads) and glue. It does not correspond to the definition of mucus which is an aqueous secretion with a gelatinous appearance that does not harden in fibers. This reference has been added and we changed the appellation to “silky tubes” to acknowledge that it is not made entirely of silk but also contains glue. “The larvae then move from the pelagic zone to the benthic zone and spin a tube made of a mix of silky fibers and glue (and not mucus, as sometimes mentioned) (Daly, 1973) in which they will stay most of their life”

6) L414: The authors describe that the number of worms per box in FF10G is different between the boxes. It is not clear to me, how it is reflected in table 2 and fig 4, and accounted for?

Therefore, the variance within a single generation (FF, poly, and traditional) is an important factor and should be included in the manuscript.

We apologize for the confusion that probably resulted from the unavailability of the actual data table (now accessible from the abovementioned link) and our report being unclear about the two sets of boxes we transplanted for the FF10 generation. One is limited to 20 worms/box and is used for comparison with the poly strain. The other set consists of boxes with different numbers of juveniles (from 12 to 30) and is used for the density test. We made it clearer in the text. 

We do not fully understand what the Reviewer #1 means by intragenerational variance. It could of course have been interesting to know how different sets of boxes of the same generation vary in median maturation age but due to obvious culture space and manpower limitations, it was not feasible. 

7) I would like to ask the authors to explain their following conclusion:

- L517: There has been no difference between the poly and poly4 generation observed, if I understood correctly. Why is the selection of 10dpf old larvae important?

The selection of 4 segment larvae at 10 dpf does not seem to make much of a difference in this particular experiment. However, batches of eggs are very variable, as mentioned in the troubleshooting guide. Many batches contain abnormally developing larvae. At a later stage, many batches contain a proportion of juveniles that have not fed at 10 dpf while being provided with algae at 4 dpf. Typically, these juveniles with empty guts are not going to develop any further and will die. Therefore, we found it important to maintain the selection of healthy larvae for the sake of optimizing each box's productivity. We have made this point clearer in the text. 

- L411-413: 'Mortality rate is nearly non-existent.' I was not able to find the corresponding data

We corrected the text to indicate that this was a general observation that we put to the test in the control of the Fig5A, which shows that for a density of up to 25 worms per box, survival at 2 mpf is above 90% then falls from 30 worms/box.

- L 376: Missing spinach in diet suggest deficit. To my knowledge, Kuehn et al. 2019 did not report problems in diet without spinach. I was wondering if there might be other factors?

We thank the reviewer for the discussion point. Kuehn et al. 2019 did not mention worm health problems. However, their own experiment (fig8) demonstrates slower growth without spinach. Both our observations and those of Kuehn et al suggest that maintaining spinach is important in a fast cycling culture. 

- L 385: It was stated, that water change was reintroduced in a few boxes – How did the authors control for this, and how is this reflected in the data?

Water change was reintroduced because we observed a few boxes with worms showing a characteristic syndrome due to water fouling (worms out of tubes, mostly immobile). Regarding the data analysis, we excluded these few boxes from the statistics of early generations of FF, to exclude biases due to the water change. 

8) The authors included some unsupported statements and speculations. I would like to ask the authors to excluded those from the text or support them with their data or a reference. I only mention some instances.

We thank the reviewer for identifying these unsupported instances. We respond point-by-point as follows:

- L 413: “growth was quick”

We integrated the text clarifying that the evidence that supports this affirmation is Figure 4, FF(6-8)G.

- L556..: paragraph about evolutionary branches

There are a large number of published results that support this paragraph: we chose Raible et al. (2005) for molecular divergence, the review Schröder et al. (2008), and Fusco and Minelli (2021) for developmental divergence in insects. We eliminated the word “morphological” as this is an idea that remains indeed very debated. These references should meet the lack of support previously stated.

- L 400: “Worm density … most important…”

To be more consistent, we decided to eliminate this sentence as it represents the conclusion rather than the preamble of this section.

- L405: “…maturation will not occur if density remains above 100 worms”

We eliminated this statement and replaced with “maturation will occur very slowly if the density remains high (our observations; Kuehn et al, 2019; Andreatta et al, 2020)

- L78-79: function of jelly in Fisher & Dorresteijn 2004 is described differently and I was not able to find citation showing decreased predation due to jelly

The article mentioned states that jelly keeps the eggs afloat after fertilization. This is never observed in the lab and it is hard to understand how the jelly which is aqueous and has the same density as water could prevent the heavier eggs to sediment. So, there is indeed no consensus on the function of the jelly. We eliminated the above statement. 

- L 378: Spirulina is very efficient for fast growing worms

We eliminated this statement.

9) The reported maturation age in the text is inconsistent (L26 vs L 588 vs 134-135), and the maturation age partially refers to earliest or median maturation in the wrong context L348-351

We do not see any inconsistency in the mentioned positions. The first mention (L26) is indeed the median maturation age whereas L588 refers to the start of the reproductive period. L134-135 refers to the traditional culture for which no published statistics is available. We thus added “our observations”. Last, L348-351 refers to the average age of the parents used for spawning the next generation (in blue in Figure 4) rather than the median maturation age (in red in Figure 4).

Suggestion: Because the food regime changed between the FF generations, I believe it is important to point out early in the results that the shorter maturation time is caused by changed feed regime, and not artificial selection.

To address this suggestion, we replace the title of the first results section with “Artificial selection did not improve the average maturation age”

10) Missing references: (only examples)

- L599 reference not included in reference list We corrected this mistake

- L357 missing reference We corrected this mistake

- L69-70 transparency good for imaging and optogenetics(?) We eliminated “optogenetics” as it is not yet published. We instead added Asadulina et al. (2012) and Özpolat et al. (2017) for whole-body and live imaging respectively.

- L 80-83: spiral cleavage 

We integrated the missing reference by quoting Chou et al. (2016).

11) L29: “… not required…an artificial moonlight regime for synchronizes sexual maturation…”

If correctly understood, the artificial moon triggers synchronized spawning, but without artificial moon light, maturation is not prevented. It was not clear to me how this is different between the traditional and new culturing method. There is no data that the new culture method affects the synchronization of maturation.

We are not sure we understood the remark pointed out here, but we tried to discuss the role of artificial moonlight. You can find the integration in lines 546-556 of the “Revised Manuscript with Track Changes” file.

12) L396 Refers to supplementary fig 2 for frozen/fresh algae, but should be Figure 5B. 

The mistake has been corrected.

13) Could the authors please clarify the following abbreviations in the text:

- L 165: NFSW

- L213/214 “w/v” and “sub-adults”

We specified in the text the above-mentioned abbreviations.

14) L 92: “The only nutrient source is the yolk contained in their macromere” I am unsure about the previous mentioned function of the lipid droplets, and I was not able to find a reference that yolk is in macromeres.

Since the yolk is distributed in all blastomeres early on, we corrected the imprecision by eliminating “in their macromeres”.

15) Could the author add the time they collected matured worms in table 2 "total matured collected? The would help to interpret especially the “classical” condition.

We added the collection time in Table 2 by specifying “total matures collected <130d”.

Minor comments:

1) Why does the feeding start at 4dpf, but Platynereis is only able to feet from 6days on?

This is an effect of raising the larvae and juveniles at 20°C. We clarified it in a new paragraph discussing the standard temperature in the discussion (from line 652 of the “Revised Manuscript with Track Changes” file). We feed juveniles as early as some of them can ingest food, again for the sake of rapid growth. 

2) The manuscript focuses on a new culturing method, and I believe a very detailed description of the life cycle, old non-functional website, comparison with other metazoan models, number of chromosomes and the 3 Rs might be irrelevant for this format and distract from the key message on speeding up the life cycle of the worms. 

We believe these paragraphs have particular importance since the obtainment of this faster strain does not end in itself, but it’s an additional step towards the 3 Rs principles that are nowadays a priority for the world of research. Additionally, the comparison with other models is fundamental to highlight the competitiveness of the FF protocol to adopt Platynereis as a reverse genetics model. For these reasons, we considered the suggestion and we decided to maintain these parts, including the description of Platynereis’s life cycle and genome.

3) The culture uses natural seawater. Is there a seasonal effect of the natural sea water onto the maturation time? 

We agree with this suggestion and we extended the conclusion paragraph debating this topic (lines 703-708 of the ”Revised Manuscript with Track Changes”).

4) L 178 and L 182: “…incubated … at 20C” Is there a reason the incubation happens at 20C, and not the standard 18C?

We discuss the issue of temperature in the relative new paragraph of the discussion. A higher temperature means faster development and maturation. However, for the sake of keeping in tune with standard table of embryonic and larval development, when using batches of eggs for actual experiment, we use a small incubator at 18°C. 

Reviewer #2: 

Major comments & suggestions:

1. lane 27-30 “A low worm density in boxes and a strictly controlled feeding has several advantages like not requiring air bubbling or an artificial moonlight regime for synchronized sexual maturation”. Q: How does the new culturing method impact hormonal pathways controlled by moonlight to promote maturation? Could there be long term effects for the health of a population of worms under these conditions? I see of course the advantage of a very efficient cycling culture system but what might be the trade offs? The authors should discuss this issue in more detail.

We added a part in the conclusion pertaining to these questions, you can find it from line 546 to line 556 of the “Revised Manuscript with Track Changes”.

2. Lane 27, 134, 355, 399 Could a concise description of the traditional protocol be included as well e.g. supplemental or direct comparison? This would make it more accessible for readers, and to gain a quicker understanding of the differences between the traditional and new culture methods.

3. Maybe it is good to create a table to compare between polymorphic worms and FF worms in the results (ex: segment, maturation, egg size, or advantage).

We agree with the need to explicit the comparison between the two strains and we thank the reviewer for the suggestion. We faced these two points together by inserting the current Table 3 in the manuscript. It represents a comparison of the food regime, culture protocol and worms’ characteristics of the FF vs Polymorphic strain.

Minor Comments:

62 Nereididae (capitalize)

Table 1: Monday, Wednesday, Friday (capitalize)

We corrected these typos.

Table 3: Addition of ascidians and sea urchins as marine invertebrates? 

We thank the reviewer for the suggestion, we added Ciona intestinalis in representation of the Ascidians, and the sea urchins Temnopleurus reevesii and Strongylocentrotus purpuratus. We also extended the list adding the stickleback Gasterosteus aculeatus.

Figure 2 Day4 Fryday (Friday)

The scheme in Figure 2 is misleading because some arrows illustrate the flow through and others the progression of the process e.g day 2 (dejellyfication) the flow through arrow is directed to 10 dpf. Maybe remove this arrow (see red circle).

We thank the reviewer for the observation concerning Figure 2. We corrected it according to the suggestion.

Figure 3: fairly low resolution figure 

The image has a resolution of 350 dpi and its dimensions are 21 x 29.7 cm. We have newly exported the figure and will upload it again to avoid upload/exportation-related problems. 

Figure 5A Density Tests or Density Test

Figure 5: A & B are outlined but C to F are not. Be consistent.

We uniformed and corrected Figure 5.

---

## [Editor Report · Decision Letter 1]

20 Nov 2023

Fast cycling culture of the annelid model Platynereis dumerilii

PONE-D-23-30265R1

Dear Dr. Balavoine,

We’re pleased to inform you that your manuscript has been judged scientifically suitable for publication and will be formally accepted for publication once it meets all outstanding technical requirements.

Kind regards,

Hector Escriva, PhD

Academic Editor

PLOS ONE
---

## [Editor Report · Acceptance letter]

4 Dec 2023

PONE-D-23-30265R1 

Fast cycling culture of the annelid model *Platynereis dumerilii*

Dear Dr. Balavoine:

I'm pleased to inform you that your manuscript has been deemed suitable for publication in PLOS ONE. Congratulations! Your manuscript is now with our production department. 

Kind regards, 

on behalf of

Dr. Hector Escriva 

Academic Editor

PLOS ONE